# On the Utility of Learning about Humans for Human-AI Coordination

**Micah Carroll**
UC Berkeley
mdc@berkeley.edu

**Rohin Shah**
UC Berkeley
rohinmshah@berkeley.edu

**Mark K. Ho**
Princeton University
mho@princeton.edu

**Thomas L. Griffiths**
Princeton University

**Sanjit A. Seshia**
UC Berkeley

**Pieter Abbeel**
UC Berkeley

**Anca Dragan**
UC Berkeley

## Abstract

While we would like agents that can coordinate with humans, current algorithms such as self-play and population-based training create agents that can coordinate with *themselves*. Agents that assume their partner to be optimal or similar to them can converge to coordination protocols that fail to understand and be understood by humans. To demonstrate this, we introduce a simple environment that requires challenging coordination, based on the popular game *Overcooked*, and learn a simple model that mimics human play. We evaluate the performance of agents trained via self-play and population-based training. These agents perform very well when paired with themselves, but when paired with our human model, they are significantly worse than agents designed to play with the human model. An experiment with a planning algorithm yields the same conclusion, though only when the human-aware planner is given the exact human model that it is playing with. A user study with real humans shows this pattern as well, though less strongly. Qualitatively, we find that the gains come from having the agent *adapt* to the human's gameplay. Given this result, we suggest several approaches for designing agents that learn about humans in order to better coordinate with them. Code is available at https://github.com/HumanCompatibleAI/overcooked_ai.

## 1 Introduction

An increasingly effective way to tackle two-player games is to train an agent to play with a set of other AI agents, often past versions of itself. This powerful approach has resulted in impressive performance against human experts in games like Go [33], Quake [20], Dota [29], and Starcraft [34].

Since the AI agents never encounter humans during training, when evaluated against human experts, they are undergoing a distributional shift. Why doesn't this cause the agents to fail? We hypothesize that it is because of the *competitive nature* of these games. Consider the canonical case of a two-player zero-sum game, as shown in Figure 1 (left). When humans play the minimizer role but take a branch in the search tree that is suboptimal, this only *increases* the maximizer's score.

However, not all settings are competitive. Arguably, one of the main goals of AI is to generate agents that *collaborate*, rather than compete, with humans. We would like agents that help people with the tasks they want to achieve, augmenting their capabilities [10, 6]. Looking at recent results, it is tempting to think that self-play-like methods extend nicely to collaboration: AI-human teams performed well in Dota [28] and Capture the Flag [20]. However, in these games, the advantage may come from the AI system's individual ability, rather than from *coordination* with humans. We claim that in general, collaboration is fundamentally different from competition, and will require us to go beyond self-play to explicitly account for *human* behavior.

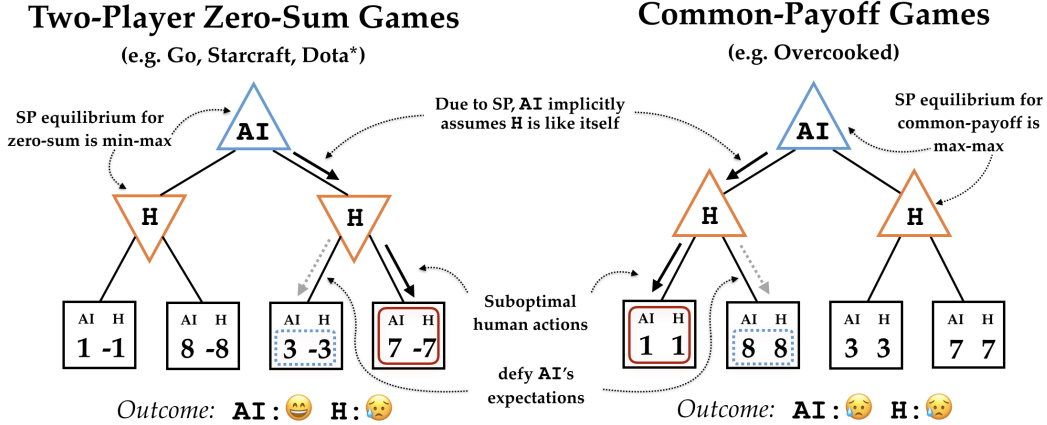

**Figure 1:** The impact of incorrect expectations of optimality. **Left:** In a competitive game, the agent plans for the worst case. **AI** expects that if it goes left, **H** will go left. So, it goes right where it expects to get 3 reward (since **H** would go left). When **H** suboptimally goes right, **AI** gets 7 reward: *more* than it expected. **Right:** In a collaborative game, **AI** expects **H** to coordinate with it to choose the best option, and so it goes left to obtain the 8 reward. However, when **H** suboptimally goes left, **AI** only gets 1 reward: the worst possible outcome!

Consider the canonical case of a common-payoff game, shown in Figure 1 (right): since both agents are maximizers, a mistake on the human's side is no longer an advantage, but an actual problem, *especially* if the agent did not anticipate it. Further, agents that are allowed to co-train might converge onto opaque coordination strategies. For instance, agents trained to play the collaborative game Hanabi learned to use the hint "red" or "yellow" to indicate that the newest card is playable, which no human would immediately understand [12]. When such an agent is paired with a human, it will execute the opaque policy, which may fail spectacularly when the human doesn't play their part.

We thus hypothesize that in true collaborative scenarios agents trained to play well with other AI agents will perform much more poorly when paired with humans. We further hypothesize that incorporating human data or models into the training process will lead to significant improvements.

To test this hypothesis, we introduce a simple environment based on the popular game *Overcooked* [13], which is specifically designed to be challenging for humans to coordinate in (Figure 2).We use this environment to compare agents trained with themselves to agents trained with a learned human model. For the former, we consider self-play [33], population-based training [19], and coupled planning with replanning. For the latter, we collect human-human data and train a behavior cloning human model; we then train a reinforcement learning and a planning agent to collaborate well with this model. We evaluate how well these agents collaborate with a held-out "simulated" human model (henceforth the "proxy human"), trained on a different data set, as well as in a user study.

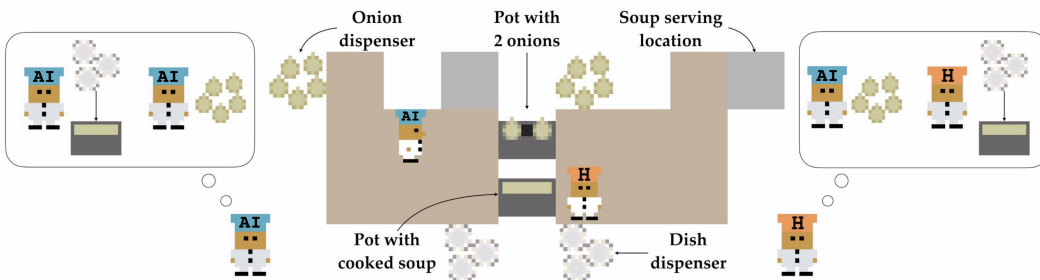

**Figure 2:** Our Overcooked environment. The goal is to place three onions in a pot (dark grey), take out the resulting soup on a plate (white) and deliver it (light grey), as many times as possible within the time limit. **H**, the human, is close to a dish dispenser and a cooked soup, and **AI**, the agent, is facing a pot that is not yet full. The optimal strategy is for **H** to put an onion in the partially full pot, and for **AI** to put the existing soup in a dish and deliver it. This is due to the layout structure, that makes **H** have an advantage in placing onions in pots, and **AI** have an advantage in delivering soups. However, we can guess that **H** plans to pick up a plate to deliver the soup. If **AI** nonetheless expects **H** to be optimal, it will expect **H** to turn around to get the onion, and will continue moving towards its own dish dispenser, leading to a *coordination failure*.

We find that the agents which did not leverage human data in training perform very well with themselves, and drastically worse when paired with the proxy human. This is not explained only by human suboptimality, because the agent also significantly underperforms a "gold standard" agent that has access to the proxy human. The agent trained with the behavior-cloned human model is drastically better, showing the benefit of having a relatively good human model. We found the same trends even when we paired these agents with real humans, for whom our model has much poorer predictive power but nonetheless helps the agent be a better collaborator. We also experimented with using behavior cloning directly for the agent's policy, and found that it also outperforms self-play-like methods, but still underperforms relative to methods that that leverage planning (with respect to the actual human model) or reinforcement learning with a proxy human model.

Overall, we learned a few important lessons in this work. First, our results showcase the importance of accounting for real human behavior during training: even using a behavior cloning model prone to failures of distributional shift seems better than treating the human as if they were optimal or similar to the agent. Second, leveraging planning or reinforcement learning to maximize the collaborative reward, again even when using such a simple human model, seems to already be better than vanilla imitation. These results are a cautionary tale against relying on self-play or vanilla imitation for collaboration, and advocate for methods that leverage models of human behavior, actively improve them, or even use them as part of a population to be trained with.

## 2   Related Work

**Human-robot interaction (HRI).** The field of human robot interaction has already embraced our main point that we shouldn't model the human as optimal. Much work focuses on achieving collaboration by planning and learning with (non-optimal) models of human behavior [26, 21, 31], as well as on specific properties of robot behavior that aid collaboration [2, 14, 9]. However, to our knowledge, ours is the first work to analyze the optimal human assumption in the context of deep reinforcement learning, and to test potential solutions such as population-based training (PBT).

Choudhury et al. [7] is particularly related to our work. It evaluates whether it is useful to learn a human model using deep learning, compared to a more structured "theory of mind" human model. We are instead evaluating how useful it is to have a human model *at all*.

**Multiagent reinforcement learning.** Deep reinforcement learning has also been applied to multiagent settings, in which multiple agents take actions in a potentially non-competitive game [24, 15]. Some work tries to teach agents collaborative behaviors [22, 18] in environments where rewards are *not* shared across agents. The Bayesian Action Decoder [12] learns communicative policies that allow two agents to collaborate, and has been applied to the cooperative card game Hanabi. However, most multiagent RL research focuses on AI-AI interaction, rather than the human-AI setting.

Lerer and Peysakhovich [23] starts from the same observation that self-play will perform badly in general-sum games, and aims to do better given some data from agents that will be evaluated on at test time (analogous to our human data). However, they assume that the test time agents have settled into an equilibrium that their agent only needs to replicate, and so they train their agent with Observational Self-Play (OSP): a combination of imitation learning and MARL. In contrast, we allow for the case where humans do not play an equilibrium strategy (because they are suboptimal), and so we only use imitation learning to create human models, and train our agent using pure RL.

**Imitation learning.** Imitation learning [1, 16] aims to train agents that mimic the policies of a *demonstrator*. In our work, we use behavior cloning [30] to imitate demonstrations collected from humans, in order to learn human models to collaborate with. However, the main focus of our work is in the design of agents that can collaborate with these models, and not the models themselves.

## 3   Preliminaries

**Multi-agent MDP.** A multiagent Markov decision process [5] is defined by a tuple $\langle S, \alpha, \{A_{i \in \alpha}\}, \mathcal{T}, R \rangle$. $S$ is a finite set of states, and $R : S \to \mathbb{R}$ is a real-valued reward function. $\alpha$ is a finite set of agents; $A_i$ is the finite set of actions available to agent $i$. $\mathcal{T} : S \times A_1 \times \cdots \times A_n \times S \to [0, 1]$ is a transition function that determines the next state given *all* of the agents' actions.

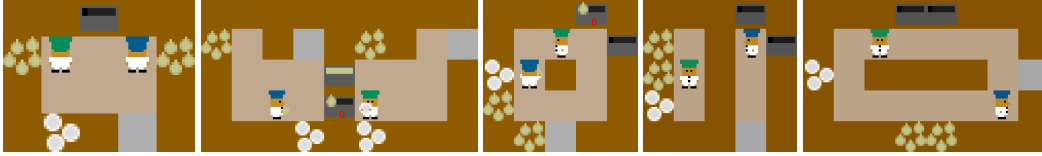

Figure 3: **Experiment layouts.** From left to right: *Cramped Room* presents low-level coordination challenges: in this shared, confined space it is very easy for the agents to collide. *Asymmetric Advantages* tests whether players can choose high-level strategies that play to their strengths, as illustrated in Figure 2. In *Coordination Ring*, players must coordinate to travel between the bottom left and top right corners of the layout. *Forced Coordination* instead removes collision coordination problems, and forces players to develop a high-level joint strategy, since neither player can serve a dish by themselves. *Counter Circuit* involves a non-obvious coordination strategy, where onions are passed over the counter to the pot, rather than being carried around.

**Behavior cloning.** One of the simplest approaches to imitation learning is given by behavior cloning, which learns a policy from expert demonstrations by directly learning a mapping from observations to actions with standard supervised learning methods [4]. Since we have a discrete action space, this is a traditional classification task. Our model takes an encoding of the state as input, and outputs a probability distribution over actions, and is trained using the standard cross-entropy loss function.

**Population Based Training.** Population Based Training (PBT) [19] is an online evolutionary algorithm which periodically adapts training hyperparameters and performs model selection. In multiagent RL, PBT maintains a population of agents, whose policies are parameterized by neural networks, and trained with a DRL algorithm. In our case, we use Proximal Policy Optimization (PPO) [32]. During each PBT iteration, pairs of agents are drawn from the population, trained for a number of timesteps, and have their performance recorded. At the end of each PBT iteration, the worst performing agents are replaced with copies of the best agents with mutated hyperparameters.

# 4   Environment and Agents

## 4.1   Overcooked

To test our hypotheses, we would like an environment in which coordination is challenging, and where deep RL algorithms work well. Existing environments have not been designed to be challenging for coordination, and so we build a new one based on the popular video game *Overcooked* [13], in which players control chefs in a kitchen to cook and serve dishes. Each dish takes several high-level actions to deliver, making strategy coordination difficult, in addition to the challenge of motion coordination.

We implement a simplified version of the environment, in which the only objects are onions, dishes, and soups (Figure 2). Players place 3 onions in a pot, leave them to cook for 20 timesteps, put the resulting soup in a dish, and serve it, giving all players a reward of 20. The six possible actions are: up, down, left, right, noop, and "interact", which does something based on the tile the player is facing, e.g. placing an onion on a counter. Each layout has one or more onion dispensers and dish dispensers, which provide an unlimited supply of onions and dishes respectively. Most of our layouts (Figure 3) were designed to lead to either low-level motion coordination challenges or high-level strategy challenges.

Agents should learn how to navigate the map, interact with objects, drop the objects off in the right locations, and finally serve completed dishes to the serving area. All the while, agents should be aware of what their partner is doing and coordinate with them effectively.

## 4.2   Human models

We created a web interface for the game, from which we were able to collect trajectories of humans playing with other humans for the layouts in Figure 3. In preliminary experiments we found that human models learned through behavior cloning performed better than ones learned with Generative Adversarial Imitation Learning (GAIL) [16], so decided to use the former throughout our experiments. To incentivize generalization in spite of the scarcity of human data, we perform behavior cloning over a manually designed featurization of the underlying game state.

For each layout we gathered ~16 human-human trajectories (for a total of 18k environment timesteps). We partition the joint trajectories into two subsets, and split each trajectory into two single-agent trajectories. For each layout and each subset, we learn a human model through behavior cloning. We treat one model, BC, as a human model we have access to, while the second model $H_{Proxy}$ is treated as the ground truth human proxy to evaluate against at test time. On the first three layouts, when paired with themselves, most of these models perform similarly to an average human. Performance is significantly lower for the last two layouts (Forced Coordination and Counter Circuit).

The learned models sometimes got "stuck": they would perform the same action over and over again (such as walking into each other), without changing the state. We added a simple rule-based mechanism to get the agents unstuck by taking a random action. For more details see Appendix A.

### 4.3 Agents designed for self-play

We consider two DRL algorithms that train agents designed for self-play, and one planning algorithm.

**DRL algorithms.** For DRL we consider PPO trained in self-play (SP) and PBT. Since PBT involves training agents to perform well with a *population* of other agents, we might expect them to be more robust to potential partners compared to agents trained via self-play, and so PBT might coordinate better with humans. For PBT, all of the agents in the population used the same network architecture.

**Planning algorithm.** Working in the simplified Overcooked environment enables us to also compute near-optimal plans for the joint planning problem of delivering dishes. This establishes a baseline for performance and coordination, and is used to perform *coupled planning with re-planning*. With coupled planning, we compute the optimal joint plan for both agents. However, rather than executing the full plan, we only execute the first action of the plan, and then *replan* the entire optimal joint plan after we see our partner's action (since it may not be the same as the action we computed for them).

To achieve this, we pre-compute optimal joint motion plans for every possible starting and desired goal locations for each agent. We then create high-level actions such as "get an onion", "serve the dish", etc. and use the motion plans to compute the cost of each action. We use $A^*$ search to find the optimal joint plan in this high-level action space. This planner does make some simplifying assumptions, detailed in Appendix E, making it near-optimal instead of optimal.

### 4.4 Agents designed for humans

We seek the simplest possible solution to having an agent that actually takes advantage of the human model. We embed our learned human model BC in the environment, treating it's choice of action as part of the dynamics. We then directly train a policy on this single-agent environment with PPO. We found empirically that the policies achieved best performance by initially training them in self-play, and then linearly annealing to training with the human model (see Appendix C).

For planning, we implemented a model-based planner that uses hierarchical $A^*$ search to act near-optimally, assuming access to the policy of the other player (see Appendix F). In order to preserve near-optimality we make our training and test human models deterministic, as dealing with stochasticity would be too computationally expensive.

## 5 Experiments in Simulation

We pair each agent with the proxy human model and evaluate the team performance.

**Independent variables.** We vary the type of agent used. We have agents trained with themselves: self-play (SP), population-based training (PBT), coupled planning (CP); agents trained with the human model BC: reinforcement learning based (PPO$_{BC}$), and planning-based P$_{BC}$; and the imitation agent BC itself. Each agent is paired with H$_{Proxy}$ in each of the layouts in Figure 3.

**Dependent measures.** As good coordination between teammates is essential to achieve high returns in this environment, we use cumulative rewards over a horizon of 400 timesteps for our agents as a proxy for coordination ability. For all DRL experiments, we report average rewards across 100 rollouts and standard errors across 5 different seeds. To aid in interpreting these measures, we also compute a "gold standard" performance by training the PPO and planning agents not with BC, but with H$_{Proxy}$ itself, essentially giving them access to the ground truth human they will be paired with.

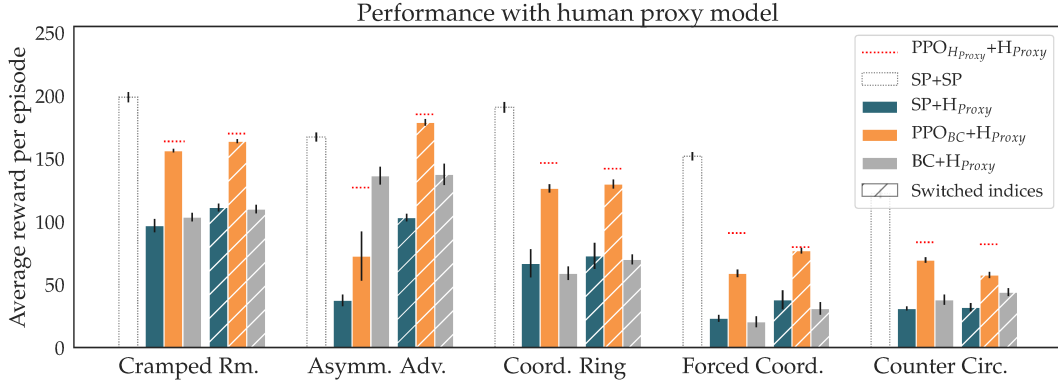

**(a) Comparison with agents trained in self-play.**

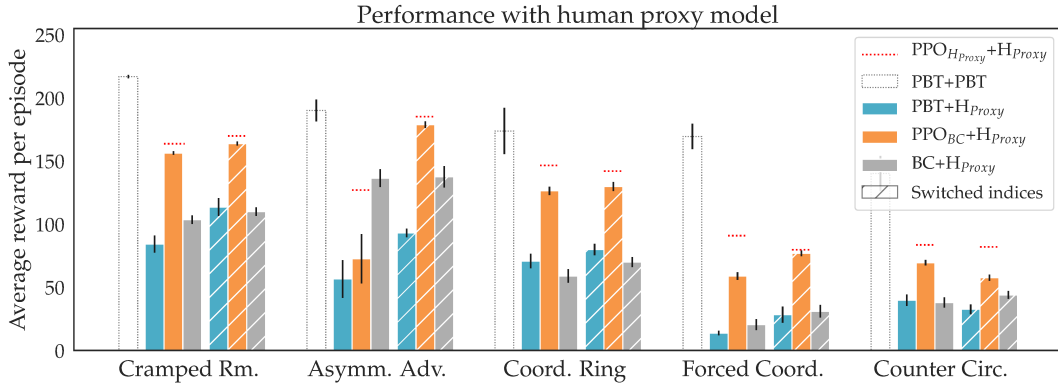

**(b) Comparison with agents trained via PBT.**

**Figure 4:** Rewards over trajectories of 400 timesteps for the different agents (agents trained with themselves – SP or PBT – in teal, agents trained with the human model – $PPO_{BC}$ – in orange, and imitation agents – BC – in gray), with standard error over 5 different seeds, paired with the proxy human $H_{Proxy}$. The white bars correspond to what the agents trained with themselves *expect* to achieve, i.e. their performance when paired with itself (SP+SP and PBT+PBT). First, these agents perform much worse with the proxy human than with themselves. Second, the PPO agent that trains with human data performs much better, as hypothesized. Third, imitation tends to perform somewhere in between the two other agents. The red dotted lines show the "gold standard" performance achieved by a PPO agent with direct access to the proxy model itself – the difference in performance between this agent and $PPO_{BC}$ stems from the innacuracy of the BC human model with respect to the actual $H_{Proxy}$. The hashed bars show results with the starting position of the agents switched. This most makes a difference for asymmetric layouts such as Asymmetric Advantages or Forced Coordination.

**Analysis.** We present quantitative results for DRL in Figure 4. Even though SP and PBT achieve excellent performance in self-play, when paired with a human model they struggle to even meet the performance of the imitation agent. There is a large gap between the imitation agent and gold standard performance. Note that the gold standard reward is lower than self-play methods paired with themselves, due to human suboptimality affecting the highest possible reward the agent can get. We then see that $PPO_{BC}$ outperforms the agents trained with themselves, getting closer to the gold standard. This supports our hypothesis that 1) self-play-like agents perform drastically worse when paired with humans, and 2) it is possible to improve performance significantly by taking the human into account. We show this holds (for these layouts) even when using an unsophisticated, behavior-cloning based model of the human.

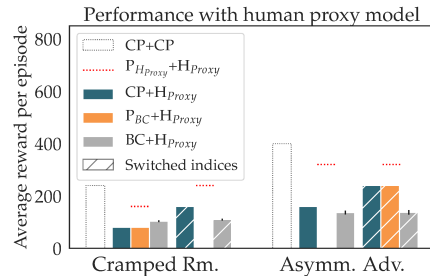

**Figure 5: Comparison across planning methods.** We see a similar trend: coupled planning (CP) performs well with itself (CP+CP) and worse with the proxy human (CP+$H_{Proxy}$). Having the correct model of the human (the dotted line) helps, but a bad model ($P_{BC}$+$H_{Proxy}$) can be much worse because agents get stuck (see Appendix G).

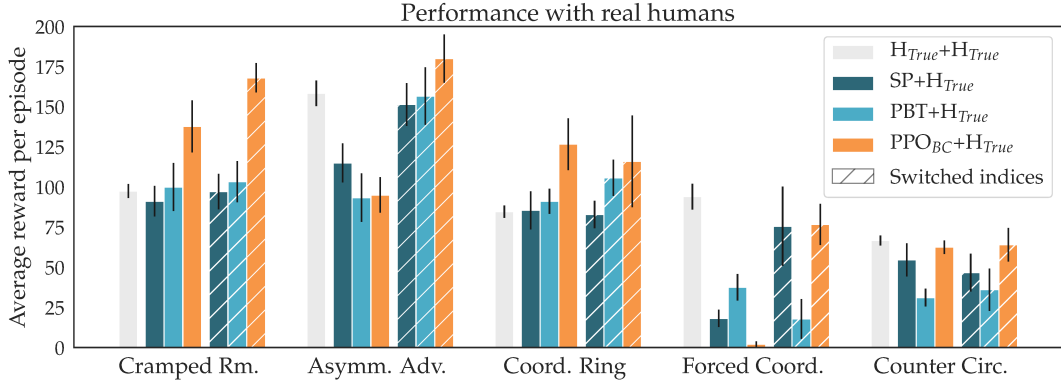

**Figure 6:** Average rewards over 400 timesteps of agents paired with real humans, with standard error across study participants. In most layouts, the PPO agent that trains with human data (PPO$_{BC}$, orange) performs better than agents that don't model the human (SP and PBT, teal), and in some layouts significantly so. We also include the performance of humans when playing with other humans (gray) for information. Note that for human-human performance, we took long trajectories and evaluated the reward obtained at 400 timesteps. In theory, humans could have performed better by optimizing for short-term reward near the end of the 400 timesteps, but we expect that this effect is small.

Due to computational complexity, model-based planning was only feasible on two layouts. Figure 5 shows the results on these layouts, demonstrating a similar trend. As expected, coupled planning achieves better self-play performance than reinforcement learning. But when pairing it with the human proxy, performance drastically drops, far below the gold standard. Qualitatively, we notice that a lot of this drop seems to happen because the agent expects optimal motion, whereas actual human play is much slower. Giving the planner access to the true human model and planning with respect to it is sufficient to improve performance (the dotted line above P$_{BC}$). However, when planning with BC but evaluating with H$_{proxy}$, the agent gets stuck in loops (see Appendix G).

Overall, these results showcase the benefit of getting access to a *good* human model – BC is in all likelihood closer to H$_{Proxy}$ than to a real human. Next, we study to what extent the benefit is still there with a poorer model, i.e. when still using BC, but this time testing on real users.

## 6  User Study

**Design.** We varied the AI agent (SP vs. PBT vs. PPO$_{BC}$) and measured the average reward per episode when the agent was paired with a human user. We recruited 60 users (38 male, 19 female, 3 other, ages 20-59) on Amazon Mechanical Turk and used a between-subjects design, meaning each user was only paired with a single AI agent. Each user played all 5 task layouts, in the same order that was used when collecting human-human data for training. See Appendix H for more information.

**Analysis.** We present results in Figure 6. PPO$_{BC}$ outperforms the self-play methods in three of the layouts, and is roughly on par with the best one of them in the other two. While the effect is not as strong as in simulation, it follows the same trend, where PPO$_{BC}$ is overall preferable.

An ANOVA with agent type as a factor and layout and player index as covariates showed a significant main effect for agent type on reward ($F(2, 224) = 6.49$, $p < .01$), and the post-hoc analysis with Tukey HSD corrections confirmed that PPO$_{BC}$ performed significantly better than SP ($p = .01$) and PBT ($p < .01$). This supports our hypothesis.

In some cases, PPO$_{BC}$ also significantly outperforms human-human performance. Since imitation learning typically cannot exceed the performance of the demonstrator it is trained on, this suggests that in these cases PPO$_{BC}$ would also outperform imitation learning.

We speculate that the differences across layouts are due to differences in the quality of BC and DRL algorithms across layouts. In Cramped Room, Coordination Ring, and the second setting of Asymmetric Advantages, we have both a good BC model as well as good DRL training, and so PPO$_{BC}$ outperforms both self-play methods and human-human performance. In the first setting of Asymmetric Advantages, the DRL training does not work very well, and the resulting policy lets the human model do most of the hard work. (In fact, in the second setting of Asymmetric Advantages in

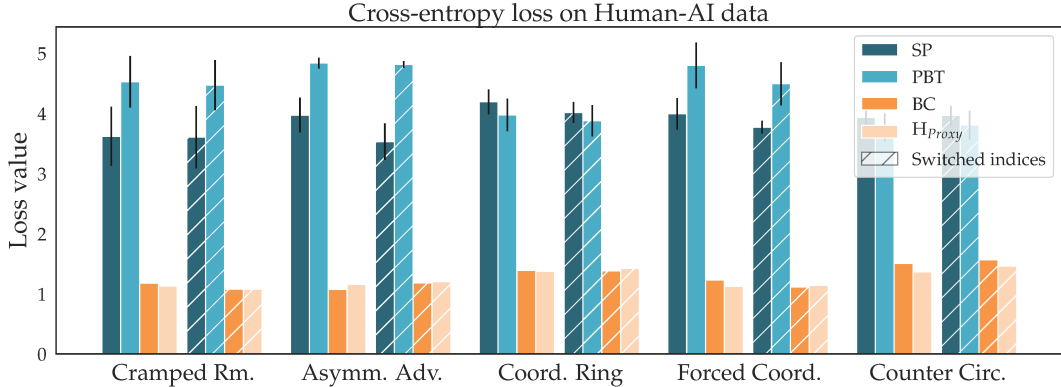

**Figure 7:** Cross-entropy loss incurred when using various models as a predictive model of the human in the human-AI data collected, with standard error over 5 different seeds. Unsurprisingly, SP and PBT are poor models of the human, while BC and $H_{Proxy}$ are good models. See Appendix H for prediction accuracy.

Figure 4a, the human-AI team *beats* the AI-AI team, suggesting that the role played by the human is hard to learn using DRL.) In Forced Coordination and Counter Circuit, BC is a very poor human model, and so PPO$_{BC}$ still has an incorrect expectation, and doesn't perform as well.

We also guess that the effects are not as strong in simulation because humans are able to adapt to agent policies and figure out how to get the agent to perform well, a feat that our simple $H_{Proxy}$ is unable to do. This primarily benefits self-play based methods, since they typically have opaque coordination policies, and doesn't help PPO$_{BC}$ as much, since there is less need to adapt to PPO$_{BC}$. We describe some particular scenarios in Section 7.

Figure 7 shows how well each model performs as a predictive model of the human, averaged across all human-AI data, and unsurprisingly finds that SP and PBT are poor models, while BC and $H_{Proxy}$ are decent. Since SP and PBT expect the other player to be like themselves, they are effectively using a bad model of the human, explaining their poor performance with real humans. PPO$_{BC}$ instead expects the other player to be BC, a much better model, explaining its superior performance.

## 7 Qualitative Findings

Here, we *speculate* on some qualitative behaviors that we observed. We found similar behaviors between simulation and real users, and SP and PBT had similar types of failures, though the specific failures were different.

**Adaptivity to the human.** We observed that over the course of training the SP agents became very specialized, and so suffered greatly from distributional shift when paired with human models and real humans. For example, in Asymmetric Advantages, the SP agents only use the top pot, and ignore the bottom one. However, humans use both pots. The SP agent ends up waiting unproductively for the human to deliver a soup from the top pot, while the human has instead decided to fill up the bottom pot. In contrast, PPO$_{BC}$ learns to use both pots, depending on the context.

**Leader/follower behavior.** In Coordination Ring, SP and PBT agents tend to be very headstrong: for any specific portion of the task, they usually expect either clockwise or counterclockwise motion, but not both. Humans have no such preference, and so the SP and PBT agents often collide with them, and keep colliding until the human gives way. The PPO$_{BC}$ agent instead can take on both leader and follower roles. If it is carrying a plate to get a soup from the pot, it will insist on following the shorter path, even if a human is in the way. On the other hand, when picking which route to carry onions to the pots, it tends to adapt to the human's choice of route.

**Adaptive humans.** Real humans learn throughout the episode to anticipate and work with the agent's particular coordination protocols. For example, in Cramped Room, after picking up a soup, SP and PBT insist upon delivering the soup via right-down-interact instead of down-right-down-interact – even when a human is in the top right corner, blocking the way. Humans can figure this out and make sure that they are not in the way. Notably, PPO$_{BC}$ *cannot* learn and take advantage of human adaptivity, because the BC model is not adaptive.

# 8 Discussion

**Summary.** While agents trained via general DRL algorithms in collaborative environments are very good at coordinating with themselves, they are not able to handle human partners well, since they have never seen humans during training. We introduced a simple environment based on the game Overcooked that is particularly well-suited for studying coordination, and demonstrated quantitatively the poor performance of such agents when paired with a learned human model, and with actual humans. Agents that were explicitly designed to work well with a human model, even in a very naive way, achieved significantly better performance. Qualitatively, we observed that agents that learned about humans were significantly more *adaptive* and able to take on both *leader and follower roles* than agents that expected their partners to be optimal (or like them).

**Limitations and future work.** An alternative hypothesis for our results is that training against BC simply forces the trained agent to be robust to a wider variety of states, since BC is more stochastic than an agent trained via self-play, but it doesn't matter whether BC models real humans or not. We do not find this likely a priori, and we did try to check this: PBT is supposed to be more robust than self-play, but still has the same issue, and planning agents are automatically robust to states, but still showed the same broad trend. Nonetheless, it is possible that DRL applied to a sufficiently wide set of states could recoup most of the lost performance. One particular experiment we would like to run is to rain a single agent that works on arbitrary layouts. Since agents would be trained on a much wider variety of states, it could be that such agents require more general coordination protocols, and self-play-like methods will be more viable since they are forced to learn the same protocols that humans would use.

In contrast, in this work, we trained separate agents for each of the layouts in Figure 3. We limited the scope of each agent because of our choice to train the simplest human model, in order to showcase the importance of human data: if a naive model is already better, then more sophisticated ones will be too. Our findings open the door to exploring such models and algorithms:

*Better human models:* Using imitation learning for the human model is prone to distributional shift that reinforcement learning will exploit. One method to alleviate this would be to add inductive bias to the human model that makes it more likely to generalize out of distribution, for example by using theory of mind [7] or shared planning [17]. However, we could also use the standard data aggregation approach, where we periodically query humans for a new human-AI dataset with the current version of the agent, and retrain the human model to correct any errors caused by distribution shift.

*Biasing population-based training towards humans:* Agents trained via PBT should be able to coordinate well with any of the agents that were present in the population during training. So, we could train multiple human models using variants of imitation learning or theory of mind, and inject these human models as agents in the population. The human models need not even be accurate, as long as in aggregate they cover the range of possible human behavior. This becomes a variant of domain randomization [3] applied to interaction with humans.

*Adapting to the human at test time:* So far we have been assuming that we must deploy a static agent at test time, but we could have our agent adapt online. One approach would be to learn multiple human models (corresponding to different humans in the dataset). At test time, we can select the most likely human model [27], and choose actions using a model-based algorithm such as model predictive control [25]. We could also use a meta-learning algorithm such as MAML [11] to learn a policy that can quickly adapt to new humans at test time.

*Humans who learn:* We modeled the human policy as stationary, preserving the Markov assumption. However, in reality humans will be learning and adapting as they play the game, which we would ideally model. We could take this into account by using recurrent architectures, or by using a more explicit model of how humans learn.

### Acknowledgments

We thank the researchers at the Center for Human Compatible AI and the Interact lab for valuable feedback. This work was supported by the Open Philanthropy Project, NSF CAREER, the NSF VeHICaL project (CNS-1545126), and National Science Foundation Graduate Research Fellowship Grant No. DGE 1752814.

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
