[Supplementary Material]

# A    Behavior cloning

We collected trajectories from Amazon Mechanical Turk. We removed trajectories that fell short of the intended human-human trajectory length of $T \approx 1200$ and very suboptimal ones (with reward roughly below to what could be achieved by one human on their own – i.e. less than 220 for Cramped Room, 280 for Asymmetric Advantages Ring, 150 for Coordination Ring, 160 for Forced Coordination, and 180 for Counter Circuit). After removal, we had 16 joint human-human trajectories for Cramped Room environment, 17 for Asymmetric Advantages, 16 for Coordination Ring, 12 for Forced Coordination, and 15 for Counter Circuit.

We divide the joint trajectories into two groups randomly, and split each joint trajectory $((s_0, (a_0^1, a_0^2), r_0), \ldots, (s_T, (a_T^1, a_T^2), r_T))$ into two single agent trajectories: $((s_0, a_0^1, r_0), \ldots, (s_T, a_T^1, r_T))$ and $((s_0, a_0^2, r_0), \ldots, (s_T, a_T^2, r_T))$. At the end of this process we have twice as many single-agent trajectories than the joint human-human trajectories we started with, for a total of approximately 36k environment timesteps for each layout.

We then use the two sets of single-agent human trajectories to train two human models, BC and $H_{Proxy}$, for each of the five layouts. We evaluate these trained behavior cloning models by pairing them with themselves averaging out reward across 100 rollouts with horizon $T = 400$.

In each subgroup, we used 85% of the data for training the behavior cloning model, and 15% for validation. To learn the policy, we used a feed-forward fully-connected neural network with 2 layers of hidden size 64. We report the hyperparameters used in Table 1. We run each experiment with 5 different seeds, leading to 5 BC and 5 $H_{Proxy}$ models for each layout. We then manually choose one BC and one $H_{Proxy}$ model based on the heuristic that the $H_{Proxy}$ model should achieve slightly higher reward than the BC model – to make our usage of $H_{Proxy}$ as a human proxy more realistic (as we would expect BC to underperform compared to the expert demonstrator).

Behavior cloning, unlike all other methods in this paper, is trained on a manually designed 64-dimensional featurization the state to incentivize learning policies that generalize well in spite of the limited amount of human-human data. Such featurization contains the relative positions to each player of: the other player, the closest onion, dish, soup, onion dispenser, dish dispenser, serving location, and pot (one for each pot state: empty, 1 onion, 2 onions, cooking, and ready). It also contains boolean values encoding the agent orientation and indicating whether the agent is adjacent to empty counters. We also include the agent's own absolute position in the layout.

To correct for a tendency of the learnt models to sometimes get stuck when performing low level action tasks, we added a hardcoded the behavior cloning model to take a random action if stuck in the same position for 3 or more consecutive timesteps. As far as we could tell, this does not significantly affect the behavior of the human models except in the intended way.

| Behavior cloning hyperparameters | | | | | |
|---|---|---|---|---|---|
| Parameter | Cramped Rm. | Asym. Adv. | Coord. Ring | Forced Co-ord. | Counter Circ. |
| Learning Rate | 1e-3 | 1e-3 | 1e-3 | 1e-3 | 1e-3 |
| # Epochs | 100 | 120 | 120 | 90 | 110 |
| Adam epsilon | 1e-8 | 1e-8 | 1e-8 | 1e-8 | 1e-8 |

**Table 1:** Hyperparameters for behavior cloning across the 5 layouts. *Adam epsilon* is the choice of the $\epsilon$ for the Adam optimizer used in these experiments.

# B    Self-play PPO

Unlike behavior cloning, PPO and other DRL methods were trained with a lossless state encoding consisting of 20 masks, each a matrix of size corresponding to the environment terrain grid size. Each mask contains information about a specific aspect of the state: the player's own position, the player's own orientation, location of dispensers of various types, location of objects on counters, etc.

In order to speed up training, we shaped the reward function to give agents some reward when placing an onion into the pot, when picking up a dish while a soup is cooking, and when picking up a soup

with a dish. The amount of reward shaping is reduced to 0 over the course of training with a linear schedule.

We parameterize the policy with a convolutional neural network with 3 convolutional layers (of sizes $5 \times 5$, $3 \times 3$, and $3 \times 3$ respectively), each of which has 25 filters, followed by 3 fully-connected layers with hidden size 32. Hyperparameters used and training curves are reported respectively in Table 2 and Figures 8.

We use 5 seeds for our experiments, with respect to which we report all our standard errors.

| PPO$_{SP}$ hyperparameters | | | | | |
|---|---|---|---|---|---|
| Parameter | Cramped Rm. | Asym. Adv. | Coord. Ring | Forced Co-ord. | Counter Circ. |
| Learning Rate | 1e-3 | 1e-3 | 6e-4 | 8e-4 | 8e-4 |
| VF coefficient | 0.5 | 0.5 | 0.5 | 0.5 | 0.5 |
| Rew. shaping horizon | 2.5e6 | 2.5e6 | 3.5e6 | 2.5e6 | 2.5e6 |
| # Minibatches | 6 | 6 | 6 | 6 | 6 |
| Minibatch size | 2000 | 2000 | 2000 | 2000 | 2000 |

**Table 2:** Hyperparameters for PPO trained purely in self-play, across the 5 layouts. For simulation, we used 30 parallel environments. Similarly to the embedded human model case, parameters common to all layouts are: entropy coefficient ($= 0.1$), gamma ($= 0.99$), lambda ($= 0.98$), clipping ($= 0.05$), maximum gradient norm ($= 0.1$), and gradient steps per minibatch per PPO step ($= 8$). For a description of the parameters, see Table 3.

**(a)** Cramped Room  **(b)** Asymmetric Advantages  **(c)** Coordination Ring

**(d)** Forced Coordination  **(e)** Counter Circuit

**Figure 8:** PPO$_{SP}$ self-play average episode rewards on each layout during training.

## C  PPO with embedded-agent environment

To train PPO with an embedded human model we use the same network structure as in the PPO$_{SP}$ case in Appendix B and similar hyperparameters, reported in Table 3. As in the PPO$_{SP}$ case, we use reward shaping and anneal it linearly throughout training.

Empirically, we found that – for most layouts – agents trained directly with competent human models to settle in local optima, never developing good game-play skills and letting the human models collect reward alone. Therefore, on all layouts except Forced Coordination, we initially train in pure self-play, and then anneal the amount of self-play linearly to zero, finally continuing training purely with the human model. We found this to improve the trained agents' performance. In Forced Coordination, both players need to learn game-playing skills in order to achieve any reward, so this problem doesn't occur.

Training curves for the training (BC) and test ($H_{Proxy}$) models are reported respectively in Figures 9 and 10.

| PPO$_{BC}$ and PPO$_{H_{Proxy}}$ hyperparameters | | | | | |
|---|---|---|---|---|---|
| Parameter | Cramped Rm. | Asym. Adv. | Coord. Ring | Forced Co-ord. | Counter Circ. |
| Learning Rate | 1e-3 | 1e-3 | 1e-3 | 1.5e-3 | 1.5e-3 |
| LR annealing factor | 3 | 3 | 1.5 | 2 | 3 |
| VF coefficient | 0.5 | 0.5 | 0.5 | 0.1 | 0.1 |
| Rew. shaping horizon | 1e6 | 6e6 | 5e6 | 4e6 | 4e6 |
| Self-play annealing | [5e5, 3e6] | [1e6, 7e6] | [2e6, 6e6] | N/A | [1e6, 4e6] |
| # Minibatches | 10 | 12 | 15 | 15 | 15 |
| Minibatch size | 1200 | 1000 | 800 | 800 | 800 |

**Table 3:** Hyperparameters for PPO trained on an embedded human model environment, across the 5 layouts. *LR annealing factor* corresponds to what factor the learning rate was annealed by linearly over the course of the training (i.e. ending at $LR_0/LR_{factor}$). *VF coefficient* is the weight to assign to the value function portion of the loss. *Reward shaping horizon* corresponds to the environment timestep in which reward shaping reaches zero, after being annealed lineraly. Of the two numbers reported for *self-play annealing*, the former refers to the environment timestep we begin to anneal from pure self-play to embedded human model training, and the latter to the timestep in which we reach pure human model embedding training. *N/A* indicates that no self-play was used during training. *# Minibatches* refers to the number of minibatches used at each PPO step, each of size *minibatch size*. For simulation, we used 30 parallel environments. Further parameters common for all layouts are: entropy coefficient ($= 0.1$), gamma ($= 0.99$), lambda ($= 0.98$), clipping ($= 0.05$), maximum gradient norm ($= 0.1$), and gradient steps per minibatch per PPO step ($= 8$). For further information, see the OpenAI baselines PPO documentation [8].

**(a)** Cramped Room

**(b)** Asymmetric Advantages

**(c)** Coordination Ring

**(d)** Forced Coordination

**(e)** Counter Circuit

**Figure 9:** PPO$_{BC}$ average episode rewards on each layout during training over 400 horizon timesteps, when pairing the agent with itself or with BC in proportion to the current self-play annealing.

# D    Population Based Training

We trained population based training using a population of 3 agents, each of which is parameterized by a neural network trained with PPO, with the same structure as in C.

During each PBT iteration, all possible pairings of the 3 agents are trained using PPO, with each agent training on a embedded single-agent MDP with the other PPO agent fixed. PBT selection was conducted by replacing the worst performing agent with a mutated version of the hyperparameters. The parameters that could be mutated are lambda (initialized $= 0.98$), clipping (initialized $= 0.05$), learning rate (initialized $= 5e - 3$), gradient steps per minibatch per PPO update (initialized $= 8$),

**(a)** Cramped Room　　　**(b)** Asymmetric Advantages　　　**(c)** Coordination Ring

**(d)** Forced Coordination　　　**(e)** Counter Circuit

**Figure 10:** $\text{PPO}_{H_{Proxy}}$ average episode rewards on each layout during training over 400 horizon timesteps, when pairing the agent with itself or with $\text{H}_{Proxy}$ in proportion to the current self-play annealing.

entropy coefficient (initialized $= 0.5$), and value function coefficient (initialized $= 0.1$). At each PBT step, each parameter had a $33\%$ chance of being mutated by either a factor of $0.75$ or $1.25$ (and clipped to the closest integer if necessary). For the lambda parameter, we mutate by $\pm\frac{\epsilon}{2}$ where $\epsilon$ is the distance to the closest of $0$ or $1$, to ensure that it will not go out of bounds.

As for the other DRL algorithms, we use reward shaping and anneal it linearly throughout training, and evaluate PBT reporting means and standard errors over 5 seeds. Hyperparameters and training reward curves are reported respectively in Table 4 and Figure 11.

| PBT hyperparameters | | | | | |
|---|---|---|---|---|---|
| Parameter | Cramped Rm. | Asym. Adv. | Coord. Ring | Forced Co-ord. | Counter Circ. |
| Learning Rate | 2e-3 | 8e-4 | 8e-4 | 3e-3 | 1e-3 |
| Rew. shaping horizon | 3e6 | 5e6 | 4e6 | 7e6 | 4e6 |
| Env. steps per agent | 8e6 | 1.1e7 | 5e6 | 8e6 | 6e6 |
| # Minibatches | 10 | 10 | 10 | 10 | 10 |
| Minibatch size | 2000 | 2000 | 2000 | 2000 | 2000 |
| PPO iter. timesteps | 40000 | 40000 | 40000 | 40000 | 40000 |

**Table 4:** Hyperparameters for PBT, across the 5 layouts. *PPO iteration timesteps* refers to the length in environment timesteps for each agent pairing training. For simulation, we used 50 parallel environments. The mutation parameters were equal across all layouts. For further description of the parameters, see Table 3.

# E   Near-optimal joint planner

As mentioned in 4.3, to perform optimal planning we pre-compute optimal joint motion plans for every possible starting and desired goal location for each agent. This enables us to quickly query the cost of each motion plan when performing $A^*$ search.

We then define the high-level actions: "get an onion", "serve the dish", etc, and map each joint high-level action onto specific joint motion plans. We then use $A^*$ search to find the optimal joint plan in this high-level action space.

This planner does make some assumptions for computational reasons, making it near-optimal instead of optimal. In order to reduce the number of required joint motion plans by a factor of 16, we only consider position-states for the players, and not their orientations. This adds the possibility of wasting one timestep when executing certain motion plans, in order to get into the correct orientation. We

**Figure 11:** Average episode rewards on each layout during training for PBT agents when paired with each other, averaged across all agents in the population.

have added a set of additional conditions to check for such a case and reduce the impact of such an approximation, but they are not general.

Another limitation of the planning logic is that counters are not considered when selecting high level action, as it would increase the runtime of $A^*$ by a large amount (since the number of actions available when holding an item would greatly increase). This is not of large importance in the two layouts we run planning experiments on. The use of counters in such scenarios is also minimal in human gameplay.

Another approximation made by the planner is only considering a 3 dish delivery look-ahead. Analyzing the rollouts, we would expect that increasing the horizon to 4 would not significantly impact the reward for the two layouts used in Section 5.

## F   Near-optimal model-based planner

Given a fixed partner model, we implement a near-optimal model-based planner that plays with the fixed partner. The planner uses a two-layered $A^*$ search in which:

**1)** on a low level we use $A^*$ in the game state space with edges being basic player joint actions (one of which is obtained by querying the partner model). To reduce the complexity of such search, we remove stochasticity from the partner model by taking the argmax probability action.

**2)** on a higher level we use $A^*$ search in the state space in which edges are high level actions, similarly to those of the near-optimal joint planner described in Appendix E. Unlike in that case, it is unfeasible to pre-compute all low-level motion plans, as each motion plan does not only depend on the beginning positions and orientations of the agents, but also on other features of the state (since they could influence the actions returned by the partner model).

## G   Planning experiments

Due to computational constrains, when evaluating planning methods (such as in Figure 5), we evaluated on a horizon of 100 timesteps, and then multiplied by 4 to make the environment horizon comparable to all other experiments. This is another source of possible suboptimality in the planning experiments.

When observing Figure 5, we see that $P_{BC}+H_{Proxy}$ performs much worse than the red dotted line (representing planning with respect to the actual test model $H_{Proxy}$). This is due to the fact that in the planning experiments all agents are set to choose actions deterministically (in order reduce the

planning complexity – as mentioned in appendix F), leading the agents to get often stuck in loops that last the whole remaining part of the trajectory, leading to little or no reward.

# H Human-AI experiments

**Figure 12:** The results are mostly similar to those in Figure 6, with the exception of larger standard errors introduced by the non-cooperative trajectories. The reported standard errors are across the human participants for each agent type.

**Figure 13:** Accuracy of various models when used to predict human behavior in all of the human-AI trajectories. The standard errors for DRL are across the 5 training seeds, while for the human models we only use 1 seed. For each seed, we perform 100 evaluation runs.

The human-AI trajectories were collected with a horizon of 400 timesteps. The agents used in this experiment were trained with slightly different hyperparameters than those in the previous appendices: after we had results in simulation and from this user study, we improved the hyperparameters and updated the simulation results, but did not rerun the user study.

After manually inspecting the collected human-AI data, we removed all broken trajectories (human not performing any actions, and trajectories shorted than intended horizon). We also noticed that in a large amount of trajectories, humans were extremely non-cooperative, not trying to perform the task well, and mainly just observing the AI agent and interacting with it (e.g. getting in its way). Our hypothesis was that these participants were trying to "push the boundaries" of the AI agents and test where they would break – as they were told that they would be paired with AI agents. We also removed these non-cooperative trajectories before obtaining our results. Cumulatively across all layouts, we removed 15, 11, and 15 trajectories of humans paired with PBT, $PPO_{SP}$, and $PPO_{BC}$ agents respectively.

In Figure 12 we report human-AI performance without these trajectories removed. Performing an ANOVA on all the data with agent type as a factor and layout and player index as a covariates showed a significant main effect for agent type on reward ($F(2, 250) = 4.10$, $p < .01$), and the post-hoc analysis with Tukey HSD corrections confirmed that $PPO_{BC}$ performed significantly better than for PBT ($p < .01$), while fell just short of statistical significance for SP ($p = .06$).

We also report the results of using various agents as predictive models of human behavior in Figure 13. This is the same setup as in Figure 7, except we are reporting accuracies here instead of cross-entropy losses.

Another thing to note is that in order to obtain Figure 7, to prevent numerical overflows in the calculation of cross-entropy for the PBT and PPO$_{SP}$ models, we lower-bounded the probability outputted by all models for the correct action to $\epsilon = 1 \times 10^{-3}$. This reduces the loss of all models, but empirically affects PBT and PPO$_{SP}$ the most, as they are the models that are most commonly confidently-incorrect in predicting the human's action. We chose this $\epsilon$ as it is about 1 order of magnitude smaller than the smallest predictions assigned to the correct actions by the human models (i.e. the worst mistakes of the human models).