[Reviews · NeurIPS 2019]

Reviewer 1



Originality: - The paper is novel in that it is the first to work with the Overcooked environment. That in itself is not worthy of high marks, but in this case what they do with it is important because they address a real problem in the current multi-agent literature around self-play being sufficient for any form of interaction. Quality: - This submission is of high quality with the proper experiments and caveats. - I am giving it really high marks because it persistently tackles the main hypothesis presented at the beginning that self-play is insufficient for collaborative environments because it does not account for humans (or, in general, any suboptimal agent) taking actions that deviate from optimal. The result is a set of experiments that deliver on the hypothesis as well as takes early steps towards addressing these issues via training alongside agents behaviorally cloned from human actions. Clarity: - The paper is well written, albeit there are typos littered here and there such as on L257 with "simulat". Significance: - These results are important, not just for what they say about self-play but very much for the environment that they introduce and the potential for research into collaborative approaches in multi-agent. I implore the authors to release this to the community and additionally ask them to consider shouting an open call for contributors to this to make it even more amenable to human interaction. This is the kind of setup that we need more of going forward.

Reviewer 2



Summary: The paper investigates the usefulness of modeling human behavior in human-ai collaborative tasks. In order to study this question, the paper introduces an experimental framework that consists of: a) modeling human behavior using imitation learning, b) training RL agents in several modes (self-play, trained agains human imitator, etc.), c) measuring the joint performance of human-AI collaboration. Using both simulation based experiments and a user study the paper showcases the importance of accounting for human behavior in designing collaborative RL agents. Comments: The topic of the paper is interesting and important for modern hybrid human-AI decision making systems. This seems like a well written paper with solid contributions: to the best of my knowledge, no prior work has systematically investigated the utility of human modeling in the context of human-AI collaboration in RL. The results are clearly presented, and the experimental study seems correct. Overall, I find the paper enjoyable to read and to be of an interest to the NeurIPS community, but I also feel that some more experimentation (larger scale user studies with a more diverse set of environments) would be beneficial. For example, it is not clear what type of human modeling would be most beneficial and for what types of tasks one might want to use human models. Nevertheless, I think this paper might be a good starting point for investigating such questions. A few questions, comments, and clarifications for the rebuttal: a) This paper seem to complement the line of work on importance sampling, that includes experimental studies of human-AI interaction in RL domains. Perhaps a good example of this line of research would be: Mandel et al., ‘Offline Evaluation of Online Reinforcement Learning Algorithms’ which argues against model-based approaches and in favor or off-policy evaluation, particularly for complex models (presumably this would include human behavior). I think it would be useful to compare the utility of human modeling vs. directly utilizing importance sampling. How would the two approaches scale with the complexity of the environment, the availability of human data, and the complexity of modeling human behavior? b) On page 6, the hypothesis test indicates that the main hypothesis is confirmed. However, in Fig. 5a only in two domains the associated error bars are not overlapping. Does the statistical test differentiate between the scenarios? Additionally, do the result of the statistical test hold for Fig. 10a in the appendix? c) I think more discussion regarding the characterization of the domains for which we obtain a substantial improvement in the joint utility would be quite valuable. From a theoretical point of view, it is not surprising that we can achieve a utility increase if we train an AI agent using a faithful human model instead of self-play (since training with a wrong type of collaborator is like having a wrong transition kernel). However, it is often not easy to obtain human trajectories (e.g., privacy, safety concerns), so it would be great to know a priori if having human model would not be beneficial. Other than that, the discussion on page 7 is quite intriguing, especially the part about leader/follower behavior. A couple of references that might be useful for this research direction: Nikolaidis et al. ‘Game-Theoretic Modeling of Human Adaptation in Human-Robot Collaboration’, Dimitrakakis et al. ‘Multi-view Decision Processes: the Helper AI Problem’. ---------- Update: I have read the author response. Thank you for clarifying some parts of the paper.

Reviewer 3



This work demonstrates how current learning algorithms perform when paired with human teammates on collaborative tasks. The authors introduce a collaborative environment based on the game Overcooked in which a team has to cook and serve meals. Current techniques that use self-play and population-based training trains agents to perform well against similar agents, but this training does not necessarily transfer well to a team that includes a human, who may not play optimally or similarly. The authors first collected human-human gameplay on the game and used a subset of the data to train a human model using behavior cloning. Results on multiple layouts of the game showed that agents trained with an approximate human model performed much better than when trained with other agents. A user study further confirmed that an agent trained with a model of the human performs better with real users. Strengths: - Incorporating models of humans into AI learning systems is an important direction for the field. - The game Overcooked can be a testbed for others interested in collaborative environments, if released to the public. The qualitative descriptions of the strategies a model learns through self-play is interesting and valuable for understanding why these models fail with humans. Weaknesses: - Demonstrating that an agent trained with a human model performs better than an agent assuming an optimal human is not necessarily a new idea and is quite well-studied in HRI and human-AI collaboration. While the work considers the idea from the perspective of techniques, such as self-play and population-based training, the authors need to justify how this is significantly different from prior work. - The idea and execution is simple. The model of the human is basic, which is fine if the idea itself is very novel, but there are many works on incorporating human models into AI learning systems. Originality: While the work is set in the context of more recent algorithms, the idea of modelling humans and not assuming humans are optimal in training is not a new concept. There are several works in a similar area, so it would be important to differentiate the work with many prior works. - Koppula, Hema S., Ashesh Jain, and Ashutosh Saxena. "Anticipatory planning for human-robot teams." Experimental Robotics. Springer, Cham, 2016. - Nikolaidis, Stefanos, et al. "Efficient model learning from joint-action demonstrations for human-robot collaborative tasks." Proceedings of the tenth annual ACM/IEEE international conference on human-robot interaction. ACM, 2015. - Freedman, Richard G., and Shlomo Zilberstein. "Integration of planning with recognition for responsive interaction using classical planners." Thirty-First AAAI Conference on Artificial Intelligence. 2017. Quality: The paper had overall high quality. The authors paid attention to details about the approach and included them in the text, which helped to understand the full procedure. It was unclear what the imitation learning condition was. Is that an agent that acts exactly as if it were a human based on the trained human BC model? If so, it seems like an inappropriate baseline since the premise of the work is that an agent is collaborating with a human rather than acting like the human acts. Clarity: The paper was written clearly. In terms of terminology: It seemed like BC and H_proxy were both trained using behavior cloning, which made the names a bit of a misnomer. In the Figure 3 caption, the hyphens made the explanation confusing. There were a few typos, included below, but overall, the approach and results were explained well. - Pg 6: taking the huaman into account → taking the human into account - Pg 6: but this times → but this time - Pg 7: simulat failure modes → similar failure modes Significance: Modelling humans when training AI systems is an important topic for the community, as many of our trained models will have to work with people while current algorithms do not always handle this. So, the general idea is definitely significant. The main concern is the originality of the work compared to prior work on modelling humans in collaborative tasks for better team performance. Other comments: - What does the planning baseline add to the story? - Was the data collection for the 5 layouts randomized? It sounds like the data was always collected in the same order, which means there may be learning effects across the different layouts. - How did you pick 400 timesteps? ----------------------- I have read the author response, and the authors make good points about how the work's contributions still provide value to the HRI and related communities. Specifically, the authors discuss the importance of considering humans in more recent deep learning frameworks and how this provides new value compared to prior works that focus on modelling humans in planning-based frameworks, which is reasonable. I additionally appreciate the experiment that the authors conduct in order to compare their method to a noisy optimality condition used in prior work.

[Author Response · NeurIPS 2019]

We would like to thank the reviewers for their thoughtful comments. Our paper makes the empirical point that state of the art deep learning techniques don't work in collaborative settings and we need human data to fix this. It also introduces the Overcooked environment as one in which coordination is key to achieve high reward. We are glad that both R1 and R2 see the significance of these contributions, and R1 is correct: we have gotten a lot of interest in the environment, and so we are planning to release code soon. However, R2 and R3 had some concerns.

**R2** *would like to see further experimentation*, and points out that one question of particular interest is when it is useful to model the human, and how that relates to a more diverse set of environments. We agree that these are two important research directions, as are many others, some of which we listed in our future work section. We see the paper as seeding many lines of inquiry, many more than we can investigate ourselves – we packed quite a bit in the paper and the appendix already. Based on initial interest, it seems that many groups will investigate these questions.

**R3** *is concerned that the HRI community has already made our empirical points*. We ask R3 to consider that *there is a difference between previous results that compared planning-based approaches and our work which emphasizes their deep learning counterparts*. We think there's value in making these points in the context of these newer techniques. There is also more nuance to our results that R3 is not giving us credit for. First, PBT is supposed to do better than pure self-play because it's designed to be robust. We find that it isn't. Second, as R2 noted, the qualitative analysis of what exactly happens when you run RL and self-play here and try it with humans is illustrative.

In addition, note that *it was not obvious a priori that self-play is insufficient*. OpenAI Five has played Dota cooperatively with humans, while For The Win agents have played Quake Capture the Flag with a human teammate (see citations in main paper). This has inspired many people to think that collaboration as a whole is addressable this way – talks about alternative methods for HRI inevitably lead to the question "what about self-play, wouldn't that solve it?"

We do think our results are of interest to the HRI community as well. Most HRI papers, including the ones R3 cites, replace an optimal human with a noisily optimal human (where data is used to learn the reward function or goal parameters). Obviously noisy rationality is a better model than perfect rationality, but we show something stronger: in our setting, *even behavior cloning does better than rationality*. This is because noisy optimality attributes mistakes to randomness (which can't be predicted), while behavior cloning can learn *how* the human makes mistakes, and so can correct for them. Our qualitative analysis shows that there are *systematic* ways in which humans deviate from optimality, and so we expect the black-box approach would work significantly better than an approach based on noisy optimality. Unfortunately, this discussion didn't make it into our paper because our target audience was different.

We did run an experiment to test this for this rebuttal. We created a simple hierarchical agent $A$, which chooses a subgoal (e.g. "get an onion"), and then chooses an action to pursue the subgoal. Both choices are modeled using noisy optimality, with $\beta$ chosen so that the behavior looks "human-like". We trained $\text{PPO}_A$ (analogous to $\text{PPO}_{BC}$) and evaluated against $H_{proxy}$. As expected, $\text{PPO}_A$ performed better than SP but worse than $\text{PPO}_{BC}$. We could likely improve $A$ with more time, but we expect it would not change the qualitative results.

We think these experiments suggest that the HRI community consider how to obtain human models that can predict *systematic* deviations, unlike noisy rationality. If nothing else, the experiments are valuable because they demonstrate that the Overcooked environment is an excellent testbed for HRI techniques.

**R2:** Thanks for the reference on the importance sampling line of work! We agree that these techniques could help. One difficulty is that human-AI gameplay visits previously unseen states as the agent learns to deal with human failures. While behavioral cloning suffers from distributional shift, importance sampling would fail entirely.

In Figure 5a, we used layouts as a covariate, but did not test interaction effects between layout and the agent type. So, the effect we got was in aggregate – it does not mean that BC helped in every layout individually. Running the test on Figure 10a produces similar but less pronounced results. There is still a significant main effect for agent type on reward ($F(2, 251) = 3.56$, $p = .03$), and the post-hoc analysis with Tukey HSD corrections shows that $\text{PPO}_{BC}$ significantly outperforms PBT ($p = .03$) and non-significantly outperforms SP ($p = .12$). We will add this to the appendix.

**R3:** The imitation learning condition is an agent that acts like a human. We included it because we have also heard (less often) the suggestion that human-AI collaboration could happen just by having the AI imitate the human.

Yes, the layout order was fixed, which could lead to learning effects. If we randomized the order, different humans would have different learning effects for the same layout, leading to higher variance results and requiring both more data to train models and more evaluation in order to detect a statistically significant effect.

With short horizons and few deliveries, policies of different skill could get the same reward. However, long horizons make DRL training harder, and might bore human players. We chose a horizon of 400 to balance these tradeoffs.

One potential explanation for our deep RL results is we didn't properly tune our self-play algorithms. We included the planning experiments to demonstrate that the problem lies squarely with the assumption of optimality, not DRL training.

[Meta-Review · NeurIPS 2019]

The paper proposes a new evaluation framework and benchmark for multi-agent learning settings where coordination with team mates is required to complete a task, and carefully evaluates state-of-the-art learning approaches in this novel setting, including evaluation with human players. All reviewers agreed that the contributions made by the paper are high, and are likely to influence future work in this field. In the initial reviews, several areas of improvement were noted, including to precisely explain the relationship of this work to the substantial amount of prior work in human-robot and human-AI interaction, several requests for clarification, and suggestions for further experimentation. The reviewers were content with the author response, and in particular the provided clarification of the relationship to prior work and overall contribution of the paper. I encourage the authors to carefully consider all reviewer comments when preparing the camera ready version.